# Opportunities of Amlodipine as a Potential Candidate in the Evaluation of Drug Compliance during Antihypertensive Therapy

**DOI:** 10.3390/medicina59020340

**Published:** 2023-02-10

**Authors:** Dmitrijs Kustovs, Inga Urtāne, Eduards Sevostjanovs, Eva Moreino, Kārlis Trušinskis

**Affiliations:** 1Department of Pharmaceutical Chemistry, Riga Stradiņš University, LV-1007 Riga, Latvia; 2Laboratory of Physical Organic Chemistry, Latvian Institute of Organic Synthesis, LV-1006 Riga, Latvia; 3Department of Internal Disease, Riga Stradiņš University, LV-1007 Riga, Latvia

**Keywords:** amlodipine, dehydro-amlodipine, adherence, hypertension, drug monitoring

## Abstract

*Background and Objectives*: Blood pressure measurement is essential evidence to establish that the chosen medicine and dosage are appropriate, and also indirectly indicates whether the medicine is being used at all. Therefore, current research compares adherence to the target blood pressure at home and in the hospital between different age groups, using similar combinations of the drugs prescribed by the doctor within ongoing antihypertensive therapy. Moreover, it is very important to develop a method for the determination of amlodipine and its metabolite, which would suitable for clinical applications, when the result is needed as quick as possible. *Materials and Methods*: This prospective study included patients aged ≥18 years who were diagnosed with hypertension. Subjects were divided into two age groups according to European Society of Cardiology (ESC) hypertension guidelines; older patients (≥65 years) and adult patients (<65 years). Assessment of adherence rate to antihypertensive medications was performed using a measurement of systolic blood pressure and comparing this to ESC hypertension guideline data. A simple liquid chromatography-tandem mass spectrometer (LC-MS/MS) method for determination of amlodipine and dehydroamlodipine was developed and validated according to the European Medicines Agency guideline on bioanalytical method validation at the Latvian Institute of Organic Synthesis. *Results*: A total of 81 patients with arterial hypertension were enrolled in this study. A significant number of patients were overweight (N = 33, 40.7%) and obese (N = 36, 44.4%). To control arterial hypertension, 70 (86.4%) patients used fixed-dose combinations, where one of the components was amlodipine. Practically, 36 (44.4%) hypertensive subjects were not able to comply with target blood pressure. Nonetheless, 38 (46.9%) patients who received fixed-dose combinations were able to comply with target blood pressure. *Conclusions*: Adherence to ESC hypertension guideline proposed target blood pressure was relatively low among hypertensive subjects even though a significant number of patients were taking fixed-dose combinations. Therefore, optimizing prevention, recognition, and care of hypertensive young adults require intensive educational interventions. Moreover, survey data suggest that therapeutic drug monitoring using the validated simple, sensitive LC-MS/MS method is pivotal for further understanding factors influencing adherence.

## 1. Introduction

The prevalence of hypertension continues to rise, making it a leading global health issue. In 2019, the prevalence of hypertension was 34% in men and 32% in women aged 30–79. Hypertension has been identified as a risk factor for the 8.5 million deaths resulting from renal diseases, ischemic heart disease, stroke, and other vascular diseases [1]. The high prevalence and mortality rates associated with hypertension have led to increased research and highly potent drugs. Despite significant advances in the diagnosis and treatment of arterial hypertension, the condition continues to pose a serious medical, social, and economic burden. Blood pressure control, which is the main goal of antihypertensive treatment, cannot be achieved with effective medication alone, but also requires cooperation with the patient [2]. Most patients require a combination of two or even three medicines to control their blood pressure within the recommended range. Therefore, such a complex antihypertensive regimen may require multiple medications to be administered several times during the day [3]. 

In turn, the use of multiple drugs and the possibility of the development of side effects reduces the patient’s adherence and causes further cardiovascular complications. 

According to the European Society of Cardiology (ESC) hypertension guideline, amlodipine (AML) often predominates as initial treatment as a single tablet or as a component of a fixed-dose combination (FDC) [4]. The drug is administered orally once daily in different dosages such as 2.5 mg, 5 mg and 10 mg tablets. The half-life of AML is 30–50 h, which is the longest compared to other dihydropyridine medications [5]. Although the many benefits of the drug, it may cause lower-extremity edema, which could make it necessary to change its dosage, become distressing to the patients or even cause patient self-managed medication discontinuation hindering adherence to the therapy. Thus, in most cases AML is administered in FDC with other antihypertensives to increase its efficacy and minimize possibility of adverse effects [6,7]. AML undergoes hepatic metabolism facilitated by cytochrome P450 enzymes (CYP). The metabolism converts AML into inactive metabolites such as a pyridine derivative. Metabolism of AML to dehydroamlodipine (DAML) is an NADPH-dependent reaction that involves the CYP3A4 enzyme [8]. Therefore, measuring the plasma concentration of DAML can help determine the successful metabolism of the drug, which correlates with its efficacy in managing high blood pressure. 

Regardless of the medication prescribed, it is important that the patient is adherent and has regular blood pressure monitoring. A low level of adherence in chronic conditions is associated with poor outcomes and an additional burden on the healthcare systems [9,10,11]. A high level of adherence is essential for the management of chronic conditions and the effectiveness of prescribed therapies. 

Furthermore, people believe that arterial blood pressure gradually increases during aging over time, hence is a normal condition of the body. However, ESC hypertension guidelines endorsed optimal blood pressure <130/80 mm Hg at the age <65 years, and 140/90 mm Hg for individuals older than 65 years. Likewise, ESC hypertension guideline outlines benefits of home blood pressure monitoring, where white-coat and masked hypertension can be identified because measurement in a home setting may be more relaxed than the doctor’s office [4]. In addition, home blood pressure monitoring has stronger prognostic significance when compared with ambulatory blood pressure measurement along with being widely available and can be integrated into a normal daily routine. However, recent data show that accuracy of home blood pressure monitoring devices remains a limiting factor, where only 30% of the devices have acceptable validation [12]. Thus, assessment of home and ambulatory blood pressure data are an important procedure to evaluate and exclude previously mentioned factors and slow down the growing burden of hypertension.

## 2. Materials and Methods

### 2.1. Study Subjects

This study was an observational, descriptive study that was conducted at Pauls Stradiņš Clinical University Hospital in Latvia. Written informed consent was obtained from all subjects prior to participation and after nature of the study was explained and all questions regarding the study were answered. Subject data were collected through a face-to-face survey during the period of 1 March 2020 to 31 June 2021. The selection criteria were: subject age over 18 years, a patient must be diagnosed with arterial hypertension (primary or secondary) independently of their risk factors, and must take AML for at least six months. The exclusion criteria was severe hepatic impairment. A questionnaire was used to collect environmental and lifestyle data. Seated blood pressure was measured three times using an automated blood pressure device (Diagnostic DM-400 IHB, Diagnosis, Białystok, Poland) following a manual procedure of blood pressure measurement. According to the 2018 ESC hypertension guidelines, it is recommended that older patients (≥65 years) and adult patients (<65 years) target blood pressure should be <140/90 mmHg and <130/80 mmHg, respectively [4]. Subjects were divided into two age groups conforming to hypertension practice guidelines. Ethical approval for this study was obtained from the Riga Stradiņš University Ethics Committee on 27 February 2020 (approval ref: 6-1/02/63). Participation in the study was voluntary. The data were collected and processed in accordance with the General Data Protection Regulation (GDPR) 2016/679 on the protection of natural persons with regard to the processing of personal data. 

### 2.2. Statistical Methods

Quantitative variables were described with arithmetical mean and standard deviation (SD) or median and the first (Q1) and third (Q3) quartiles, if data were not normally distributed. The independent samples *t*-test was used to test the difference in age between gender, as data were normally distributed and homogenous as assessed using the Shapiro-Wilk test with normal Q-Q plots and Levene’s test, respectively. The difference of dose of amlodipine between age groups was tested using the Mann-Whitney U test, because data were not normally distributed. Categorical or qualitative variables were characterized as number and percentage. Categorical variables were compared with Pearson χ2 test or Fisher exact test, depending on the violation or satisfaction of the assumption. The statistical analyzes were performed using SPSS 26 (IBM, Chicago, IL, USA) and G*Power 3.1.9. software (Heinrich Heine Universität Düsseldorf, Düsseldorf, Germany). Differences were considered statistically significant at *p* < 0.05. 

### 2.3. Method Validation

The analytical method was developed and validated according to the European Medicines Agency (EMA) guideline on bioanalytical method validation [13]. Validation of AML and DAML is described in Appendix A. 

## 3. Results

### 3.1. Results from Participant Surveys

A total of 81 patients with hypertension were enrolled in this study. The core task was to establish a target population of patients with possible changes in drug compliance despite the wide range of FDC combinations and in whom it would be useful to determine the concentration of amlodipine in the blood in the case of continuous use of the antihypertensive medication. The mean age of the population was 66.6 ± 9.1 years between 45 to 87. The average age difference between the genders was statistically significant—for men 64.6 ± 9.2 years and women 69.8 ± 8.0 years (t = 2.604, df = 79, *p* = 0.011, d = 0.60 (medium effect size), power = 0.86). All subjects had been taking AML for at least six months. Characteristics of participants and their AML therapy are shown in Table 1.

To control arterial blood pressure, 40 (49.4%) and 30 (37.0%) patients used two and three-drug antihypertensive FDCs, where one of the active pharmaceutical ingredients was amlodipine, more often at a dose of 5 mg. Despite widespread use of FDCs, the average dose of amlodipine in patients over 65 years of age was higher; in patients under 65 years of age the dose of amlodipine was 5.4 ± 2.3 mg (median = 5, Q1–Q3 5–5) and in patients over 65 years of age the dose of amlodipine was 6.9 ± 2.6 mg (median = 5, Q1–Q3 5–10) (U = 555, *p* = 0.005, r = 0.32 (medium effect size), power = 0.86). In order to evaluate answers regarding the regular use of the medicine, the average blood pressure values at home were compared to patient’s measured blood pressure in the hospital (see Table 2).

Findings showed that 32 (39.5%) subjects among all 36 (44.4%) hypertensive patients, were receiving FDC antihypertensive medications, where one of the components was AML. Most of the patients, 69 (85.2%) were overweight or obese. According to hospital and home systolic blood pressure measurements, patients in age group <65 years (hospital—52.8%, home—54.8%) were more careless about their health, and less compliant to target blood pressure as patients in ≥65 years (hospital—37.8%, home—26.7%) group. Furthermore, patients in age group ≥65 years were more often reaching target diastolic blood pressure (hospital—80.0%, home—97.7%). In addition, 16 (19.8%) patients confirmed that used to forget to take their medicines as prescribed by a doctor. In Figure 1 patients were divided into two groups: <65 years (a) and ≥65 years (b) according to gender. Each group was split into two subgroups according to hypertension guideline recommendations for target blood pressure values.

Data in Figure 1 represents that men without regard to age group, <65 or ≥65 years, have more often failed to reach their target blood pressure, 14 and 9, respectively. Furthermore, men in both age groups (N = 22, 27.2%) self-reported that they used to forget to take their antihypertensive medications more often compared to women (N = 9, 11.1%). Data in Figure 2 represent that most commonly seen pharmacological groups in double and triple FDCs in both age groups <65 or ≥65 years, are ACE-inhibitors and diuretics agents.

### 3.2. Results of AML and DAML Validation

Results of AML and DAML validation described in Appendix A.

## 4. Discussion

As comorbidities and the number of simultaneously used drugs increases with age, FDC is a major solution to improve compliance during arterial hypertension therapy and a tool to reach target blood pressure. Despite the simplified treatment regimen using FDCs, however, individual patient factors tend to influence the use of the drug. Even so, Shuangjiao et al. reported that medication literacy has a positive relationship with medication adherence. Therefore, adequate knowledge obtained from clinical professionals as well as community pharmacists, would improve patient attitude, behavior and adherence to medication therapy toward hypertension treatment [14].

Comparing the results with other studies will help understand the position of different authors regarding the adherence to target systolic blood pressure by comparing the variables such as age or age group, gender, and risks such as being overweight or obese. 

Various authors have explored the relationship between adherence to antihypertensive medication and subgroups. For instance, the results of the current study paralleled Fleig et al., where the authors explored the effect of FDC of perindopril or amlodipine among patients with arterial hypertension. The researchers demonstrated that 80.6% of the patients treated with FCDs of perindopril/amlodipine were either overweight (BMI >25 and <30 kg/m^2^, 46.8%) or obese (BMI >30 kg/m^2^, 33.8%). Studies recorded similar patient characteristics when compared to the current study, which recorded 85.2% obese cases. Fleig et al. demonstrated that patient age could not influence blood pressure response. The outcome established that 70.6% of patients <65 years attained blood pressure values of <140/90 mm Hg compared to 66.3% among patients ≥65 years [15]. Similar results were also observed in the study by Pallangyo et al. that showed groups of ≤60 vs. >60 years displaying similar adherence of 76.8% vs. 77.1%, respectively [16]. This is inconsistent with the current research that showed that most of the patients aged 65 years reached their SBP target compared to the subgroup of <65 years. In this study, the sub-group of <65 years of age recorded a significant decrease in blood pressure, and a similar outcome was observed in patients 65 years and older. Finally, the authors assessed adherence to medication and found that 47.2% of patients showed perfect adherence to treatment, and adherence increased by 20.6% among previously treated patients, while 51.7% of patients without antihypertensive treatment showed perfect adherence [15]. Verma et al. conducted an on-treatment analysis to determine adherence among FDC therapy and multi-pill combination therapy and found no statistical difference in primary and secondary outcomes [17]. However, the intention-to-treat analysis demonstrated high adherence among the FDC group compared to the multi-pill group (70% vs. 42% of total days). Interestingly, other studies have revealed similar results compared to the current study. Choi et al. predicted medical adherence using regression analysis and established that subgroups of ≥65 years who were treated with antihypertensive drugs showed good adherence compared to the <65 years subgroup [18]. In another study, Sheppard et al. measured adherence to medications by testing the urine samples and conducting a statistical analysis to determine whether the blood pressure was controlled at <140/90 mm Hg among patients in the ≥ 65 years of age subgroup. Results from this study showed that a total of 182 participants, or 95.3% to 97.8%, were fully adherent to all of their antihypertensive medications [19]. Non-adherence to antihypertensive mediation was generally low. Therefore, examining the relationship between the sub-groups and the desire to reach the target blood pressure revealed varied results compared to the current study. 

Furthermore, Uchmanowicz et al. established that adherence to antihypertensive medications was also driven by other factors such as gender, marital status, and educational level. Thus, the authors utilized the Hill-Bone scale in testing the levels of adherence and showed that patients obtained an average of 12.05 points per question, corresponding to 1.34 points per question [20]. Males raised the 1.34 points compared to females. However, the results differed from the current study, which revealed that men often failed to reach their target SBP. Furthermore, Consolazio et al. demonstrated significant gender differences in the youngest age group, which showed that women appeared to be less commonly treated compared to men. However, women ≥ 65 years of age were treated more often than men [21]. Without considering the sub-group factor (<65 or ≥ 65 years), men recorded a significantly higher reduction in the SBP, while no significant reduction was observed in women. In other words, men were more compliant with antihypertensive therapy compared to women [22]. Conversely, Pallangyo et al. documented that males recorded similar adherence to females (75.6 vs. 77.6%) [16]. Furthermore, patients treated in primary care settings showed a significant reduction in SBP compared to patients treated in secondary and tertiary care settings. Hence, the current study agrees partially with the results from other studies. While the current study disregards the role of subgroups, previous studies have revealed an important connection between age and gender.

The relationship between the challenge of forgetting and adherence to medications has been explored in various studies. In a qualitative study by Mostafavi et al. examining the barriers to medication adherence, the researchers developed four important themes, including the incompatibility of patients, environmental challenges of life, forgetting to take medications, and inefficient recommendations of family. The study demonstrated that forgetfulness was a barrier to medication adherence, and the challenge often occurs in the early stages of the disease [23]. Forgetfulness commonly occurs among patients taking only antihypertensive medications and at the onset of the diagnosis stage. Furthermore, Gavrilova et al. examined the adherence level to arterial hypertension and established that non-adherence to drug therapy occurred among 45.9% of the respondents. Thirty-six persons un-knowingly used the drug incorrectly, while 42 did it intentionally [24]. It was also established that the lowest adherence was recorded in patients taking medications for 2–4.9 years, but the adherence rate increased with increased hospitalization episodes. Differences in group adherence were controlled by factors such as net income, medical co-payment, employment, and the cost of treatment. Forgetfulness was also recorded among 38.3% of the patients and was considered a barrier to adherence [16]. While the current study established forgetfulness as a barrier to adherence, Gavrilova et al. found that some patients did it intentionally. 

Due to the fact that this study was conducted during the COVID-19 era, it is important to examine how adherence changed during this period. Shimels et al. examined the magnitude of poor medication adherence among hypertensive and diabetic patients during the COVID-19 period. Shimels et al. noted that the magnitude of adherence to drug therapy was at 72%, when the patients failed to meet all the requirements [25]. Nearly 57% of the patients reported the impact of COVID-19 on the availability of medications, affordability, and follow-up visits. A cross-sectional study conducted during the COVID-19 pandemic established that the level of poor antihypertensive medication adherence was 63% and connected to several factors. Factors associated with poor medication adherence during COVID-19 were lack of formal education, existing comorbid conditions, poor knowledge about hyper-tension, poor patient-physician relationships, and unavailability of medication [26]. A study conducted in Turkey to assess blood pressure management and adherence to medication established that 58.7% had their blood pressure checked during the pandemic, while 15.5% changed their dosage or timing without doctors’ approval [27]. In the study, 37.9% of patients were largely affected by the pandemic. Changes in medical adherence were impacted by failure to seek check-up services, fear of the pandemic, and impact on health.

In terms of clinical application, analysis of drug concentrations in blood serum is useful in detecting patient nonadherence to antihypertensive medication. Many recent studies found high rates of non-adherence to antihypertensive therapies which prompt simple and reliable monitoring of medication adherence with special attention to at-risk groups of patients [28,29,30,31,32]. Meantime, AML is the perfect first-line medicine available in a single fixed-dose combination form for different cardiovascular diseases with excellent pharmacokinetic parameters [33,34]. The large half-life period is perfect for TDM and assessment of adherence [35,36,37]. Regarding the assessment of amlodipine concentration in the serum, a number of methods to determine AML alone or in combination with other medications using LC-MS have been reported, but there is still scope for improvement [38,39,40,41,42,43,44,45]. Moreover, they do not detect DAML which is the principal metabolite in the AML metabolic dehydrogenation pathway. Quantification of the DAML using LC-MS/MS is crucial as it allows to understand metabolic rate of AML without implementing a separate analysis for the determination of cytochrome P450 metabolic activity. It is valuable to assess CYP metabolic activity during the pharmacotherapy, because CYP isozymes are involved in clinically important drug metabolisms. However, assessment of CYP metabolic activity is a time-consuming process in routine practice and inter-assay variability can lead to misinterpretation of the results [46,47]. Thus, LC-MS/MS analysis determining presence of the parent drug, its metabolite and both compound concentrations would be more informative in contrast to CYP metabolic activity process alone.

In the present manuscript, we have expanded the linearity range, which is approximately two times higher than those of most LC-MS/MS methods [39,40,42,44,45]. Improved dynamic range is an essential part of the quantification of drug response during personalized therapy. Moreover, sample preparation requires only 50 μL plasma, which is less than that in the published methods [39,40,41,42,44,45,48]; prepared using protein precipitation using a mixture of methanol-acetonitrile (3:1, *v*/*v*). Therefore, protein precipitation in the current procedure is low cost, requires minimal equipment, and a large number of samples can be prepared in a short time and the time or labor per sample is low during the daily routine.

## 5. Conclusions

Overall, the outcome of the current study established that nearby 50% of subjects did not achieve target values of systolic blood pressure and 85% of patients were overweight or obese. Moreover, the results agree partially with the results from other studies regarding adherence to antihypertensive medications. However, the current study disregards the role of subgroups, while previous studies have revealed an important connection between age and gender. Lastly, the age group <65 years reached target SBP and DBP in the hospital more often than at home, which was contrary to the patients in the age group ≥ 65 years. Moreover, further quantification of AML and its metabolite in patient plasma using the proposed simple, timesaving, highly sensitive validated LC-MS/MS method would help to unveil efficacy for improving blood pressure and patient outcomes depending on dosage among patients with elevated BMI, as well as affirm frequency of use of antihypertensive medication and uncover if the drug is not being used at all.

## Figures and Tables

**Figure 1 medicina-59-00340-f001:**
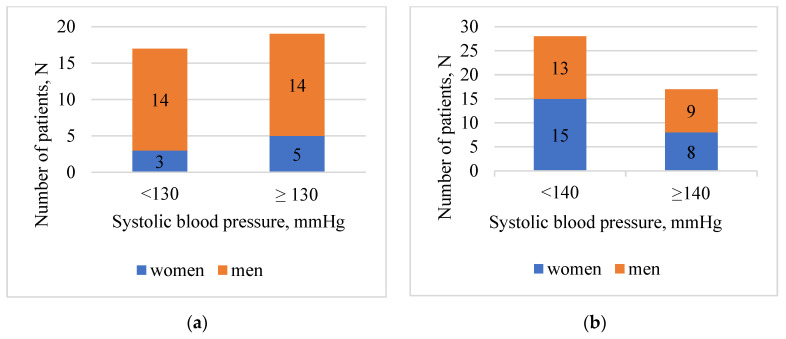
Hospital systolic blood pressure depending on the target blood pressure in the relevant age group. (**a**) Age group <65 years. (**b**) Age group ≥65 years.

**Figure 2 medicina-59-00340-f002:**
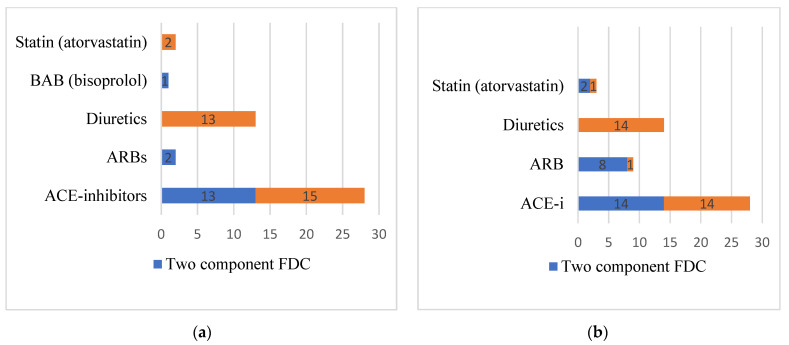
Pharmacological groups used by patients in FDC with the amlodipine. (**a**) Age group <65 years. (**b**) Age group ≥65 years.

**Table 1 medicina-59-00340-t001:** Characteristics of study participants.

Characteristics	Value
Women, *n* (%)	31 (38.7)
Men, *n* (%)	50 (61.7)
Mean age, years ± SD:	66.6 ± 9.1
women	69.8 ± 8.0
men	64.6 ± 9.2
Antihypertensive drug combination containing AML, *n* (%)	
single	11 (13.6)
double	40 (49.4)
triple	30 (37.0)
Dose of AML (mg), *n* (%)	
2.5	7 (8.6)
5	50 (61.7)
10	24 (30.9)

**Table 2 medicina-59-00340-t002:** Systolic (SBP), Diastolic (DBP) blood pressure at home and in hospital, body mass index (BMI) and medication use habits according to the age group.

Results			Age Group	
			<65 Years (%)	≥65 Years (%)
Total			36	45
Female			8 (22.2)	23 (51.1)
Male			28 (77.8)	22 (48.9)
BMI, kg/m^2^	Women	18.5–24.99	0 (0)	1 (2.2)
		>25–29.99	4 (11.1)	10 (22.2)
		≥30	4 (11.1)	12 (26.7)
	Men	18.5–24.99	5 (13.9)	6 (13.3)
		>25–29.99	11 (30.6)	8 (2.2)
		≥30	12 (33.3)	8 (17.8)
Respondents that reached target SBP (measurement in hospital)			17 (47.2)	28 (62.2)
Respondents that reached target SBP (measurement at home *)			14 (38.8)	32 (71.1)
Respondents that reached target DBP (measurement in hospital)			14 (38.9)	36 (80.0)
Respondents that reached target DBP (measurement at home *)			11 (35.5)	43 (97.7)
AML formulations of patients with reached target SBP (<130 mm Hg at the age <65 years; <140 mm Hg at the age ≥65)	Single		3 (8.3)	4 (8.9)
	Two component FDC		7 (19.4)	17 (37.8)
	Three component FDC		7 (19.4)	7 (15.6)
Respondents that failed to reach target SYS blood pressure (measurement in hospital)			19 (52.8)	17 (37.8)
Respondents that failed to reach target SYS blood pressure (measurement at home *)			17 (54.8)	12 (26.7)
Respondents that failed to reach target DBP (measurement in hospital)			22 (61.1)	9 (20.0)
Respondents that failed to reach target DBP (measurement at home *)			20 (64.5)	1 (2.3)
AML formulations of patients with non-reached target SBP (>130 mm Hg at the age <65 years; >140 mm Hg at the age ≥65)	Single		2 (5.6)	2 (4.4)
	Two component FDC		9 (25.0)	7 (15.6)
	Three component FDC		8 (22.2)	8 (17.8)
Respondents that forget to take hypertension medications	Women		2 (5.6)	7 (15.6)
	Men		14 (38.9)	8 (17.8)

* Five respondents (women—2, men—3) in <65 and one respondent (women—1) in ≥65 years group were not able to verify their home blood pressure measurements.

## Data Availability

Data sharing not applicable.

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
