# Peer review of "Opportunities of Amlodipine as a Potential Candidate in the Evaluation of Drug Compliance during Antihypertensive Therapy"

_medicina, 2023, doi:10.3390/medicina59020340_

Round 1

Reviewer 1 Report (Previous Reviewer 3)

The article Opportunities of amlodipine as potential candidate in the evaluation of drug compliance during antihypertensive therapy  is very interesting. The authors added a usefull information.

This is an usefull study. Abstract is well-presented, usage of abbreviations are good. Introduction section is good and explained in an easily understandable manner. Discussion part is well-presented. This will be useful to the young researchers. Figures presentation is clear and easily understandable. Given references are appropriate and useful supports the data presented in the article.

Author Response

Dear Reviewer,

We appreciate you sending us your feedback. However, we made minor corrections according to another reviewer comments.

Please find enclosed cover letter that indicates all corrections we made.

Thank you for your consideration of this manuscript!

Sincerely,

Dmitrijs Kustovs

Reviewer 2 Report (New Reviewer)

Dear Author(s),

Thank you for you manuscript entitled 'Opportunities of amlodipine as a potential candidate in the evaluation of drug compliance during antihypertensive therapy'. Evaluation of chronic pharmacotherapy compliance is very important topic, especially when connected with diseases, as arterial hypertension, that are connected with a wide spectrum of complications.

Methodology of your manuscript is adequate, results are presented clerarly (however I have concerns regarding some statistical analyses enlisted below), conclusion section can be more precised and more based on the currently obtained results; whereas introduction section is to extensive.

Thus, I have only minor suggestions to disclose:

i) Introduction is possibly to extensive

ii) How did you determine the sample size? Please provide power analysis data.

iii) For quantitative variables you should check the normality of distribution with KS test in order to check if mean (+-SD) or median (min-max) should be used for data presentation. Latter is also detrimental to check distribution due to distinguishing between parametric or non-parametric test usage (e.g. for comparison of age between male and female group as well for comparison of average amlodipine dose between >65 vs <65 yr. of age group (t-test vs Mann-Whitney test ...).

iv) provide a little bit more info regarding method for the determination of amlodipine and its metabolite within the materials and methods section of your abstract.

v) conclusion need to be summarized and based only on results obtained in the present study; leave assumptions for discussion section only.

vi) Do you maybe have results regarding the association between AH duration / chronic antihypertensive pharmacotherapy duration and the level of compliance/adherence? Please provide or elaborate. 

Author Response

Dear Reviewer,

We appreciate you sending us your feedback. 

Please find enclosed cover letter that indicates all corrections we made according to your comments.

Thank you for your consideration of this manuscript!

Sincerely,

Dmitrijs Kustovs

This manuscript is a resubmission of an earlier submission. The following is a list of the peer review reports and author responses from that submission.

Round 1

Reviewer 1 Report

I had the opportunity to review the paper entitled "Opportunities of amlodipine as a potential candidate in the evaluation of drug compliance during antihypertensive therapy". 

The statistical analysis and the provided results are not associated with the reported aim. The results show the differences between two group population (devided based on age) and also decribes concetrations of amlodipine. These results are enough neither to cover the aim nor to show a possible direction regarding amlodipine and (non-)adherence.

The patients are on fixed doses of amlodipine, so if the amlodipine concentraton is low this doesnt mean that the patient does not receive the drug due to amlodipine but it might be due to the other anti-hypertensive drug classes.

Also, there is no report about the % doses of the fixed compinations, % of  no reported statistical significant levels for the comparisons, not available quastionnaire to identify any possible reason of non-adherence (for example cough, oidema, polyouria). Number of patients too low. 

Author Response

Dear Reviewer,

Thank you for the review and the suggestions. We have made corrections to the article and hope that you will find the improved version more fitting for publication.  We have made following corrections:

  • information about the other drugs used by patients in fixed-dose combination with the amlodipine has been added;
  • information regarding diastolic blood pressure values in hospital and home has been added;
  • exclusion criteria (that can impact amlodipines' concentration) were entered in section 2. Materials and Methods. 2.1. Study subjects.

Question regarding available questionnaire to identify any possible reason of non-adherence (for example cough, oedema, polyuria) - unfortunately, we do not have such objective data. Patients haven't reported any notable adverse reactions, such as dry cough to ACEIs, excessive micturition to diuretics during face-to-face survey.

Unfortunately, in our study we have lack of statistical power due to a low number of participants (n = 81) likely contributed to the inability to find statistical significance for all available parameters. During the pandemic, the recruitment of participants to non-COVID-19-related clinical studies has been negatively impacted by issues including prioritisation of COVID-19 research, redeployment of research staff and the need for social distancing. Anxieties relating to the pandemic have been elevated amongst the hypertensive patient community, particularly in relation to infection risk due to self-isolation and difficulty accessing usual care. Current study conducted during the COVID-19 era, where attendance at the study site was possible, however, participants faced the additional risk of infection from contact with other participants or medical staff.

Due to an overwhelming number of COVID-19 clinical trials, we managed to assess and summarize the need to determine the concentration in case of failing to reach the target blood pressure merely in limited number of participants.

We thank you for your advice on our manuscript and would like to resubmit our revised manuscript entitled “Opportunities of amlodipine as a potential candidate in the evaluation of drug compliance during antihypertensive therapy” for further consideration.

Sincerely,  

Dmitrijs Kustovs

Reviewer 2 Report

The idea is good, the topic important, and methodology should be analyzed by an expert in HPLC. 

Author Response

Dear reviewer, 

thank you for the review and the suggestions. We hope that there will be an expert in HPLC participating in reviewing as well.

Reviewer 3 Report

The manuscript Opportunities of amlodipine as a potential candidate in the evaluation of drug compliance during antihypertensive therapy is a pivotal study, that prove mostly a scientific, but a little bit less clinical, point of view to patient’s adherence to antihypertension treatment. 

The whole pharmacological part is written in an excellent way, a precise methodology was used and described in detail, validated by European Medicines Agency (EMA).

I have some suggestions to authors:

It would be useful to explain the shorcut also in the Abstract – AML, DAML.

I would strongly recommend the authors to provide a further information about the other drugs used by patients in fixed-dose combination with the amlodipine. (f.e. authors also use a citation where are these information provided – n 21 – Fleig et al.)

The other important information should be provided about the diastolic blood pressure. These values are not mentioned in this study and this fact is strongly missing following the the European Society of Hypertension Guidelines point of you. The authors use terminology in the Abstract part - Systolic blood pressure adherence rate, but should be explained in more detail.

My next suggestion to authors is to add and information about the patients group, who and why were excluded, because in the hypertensive population are many factors, that can influence the blood pressure (f.e. was the secondary hypertension excluded?).

The discussion part is written concisely and in a clear way, the COVID - 19 era influenced also this study. I have one suggestion to improve in this part. The citaion 21 is used twice, but on the contrary  – the lines 270-275 are absolutely clear. But then in the next section of the text, the lines 276-278 (However, Fleig, et al...)

This author, Mr. Fleig also demonstrated the fact, that the patient age could not influence blood pressure responce (line n 276).

Author Response

Dear Reviewer,

Thank you for the review and the suggestions. We have made corrections to the article and hope that you will find the improved version more fitting for publication. We have made following corrections:

  • shortcuts (AML, DAML) in the Abstract part have been explained;
  • information about the other drugs used by patients in fixed-dose combination with the amlodipine has been added;
  • information regarding diastolic blood pressure values in hospital and home has been added;
  • "Systolic blood pressure adherence rate" in the Abstract part has been corrected;
  • exclusion criteria were entered in section 2. Materials and Methods. 2.1. Study subjects;
  • corrections were made regarding citation 21 where it was used twice.

Unfortunately, in our study we have lack of statistical power due to a low number of participants (n = 81) likely contributed to the inability to find statistical significance for all available parameters. During the pandemic, the recruitment of participants to non-COVID-19-related clinical studies has been negatively impacted by issues including prioritisation of COVID-19 research, redeployment of research staff and the need for social distancing. Anxieties relating to the pandemic have been elevated amongst the hypertensive patient community, particularly in relation to infection risk due to self-isolation and difficulty accessing usual care. Current study conducted during the COVID-19 era, where attendance at the study site was possible, however, participants faced the additional risk of infection from contact with other participants or medical staff.

Due to an overwhelming number of COVID-19 clinical trials, we managed to assess and summarize the need to determine the concentration in case of failing to reach the target blood pressure merely in limited number of participants.

We thank you for your advice on our manuscript and would like to resubmit our revised manuscript entitled “Opportunities of amlodipine as a potential candidate in the evaluation of drug compliance during antihypertensive therapy” for further consideration.

Sincerely,  

Dmitrijs Kustovs